# Spray-On Liquid-Metal Electrodes for Graphene Field-Effect Transistors

**DOI:** 10.3390/mi10010054

**Published:** 2019-01-14

**Authors:** Jordan L. Melcher, Kareem S. Elassy, Richard C. Ordonez, Cody Hayashi, Aaron T. Ohta, David Garmire

**Affiliations:** 1Department of Electrical Engineering, University of Hawaii at Manoa, Honolulu, HI 96822, USA; elassy@hawaii.edu (K.S.E.); aohta@hawaii.edu (A.T.O.); david.garmire@gmail.com (D.G.); 2Space and Naval Warfare Systems Center Pacific, Pearl City, HI 96782, USA; richard.c.ordonez@navy.mil (R.C.O.); cody.hayashi@navy.mil (C.H.)

**Keywords:** graphene, Galinstan, Liquid-Metal, spray-on, aerosol, honey, mobility, contact resistance, TLM, I-V characteristics

## Abstract

Advancements in flexible circuit interconnects are critical for widespread adoption of flexible electronics. Non-toxic liquid-metals offer a viable solution for flexible electrodes due to deformability and low bulk resistivity. However, fabrication processes utilizing liquid-metals suffer from high complexity, low throughput, and significant production cost. Our team utilized an inexpensive spray-on stencil technique to deposit liquid-metal Galinstan electrodes in top-gated graphene field-effect transistors (GFETs). The electrode stencils were patterned using an automated vinyl cutter and positioned directly onto chemical vapor deposition (CVD) graphene transferred to polyethylene terephthalate (PET) substrates. Our spray-on method exhibited a throughput of 28 transistors in under five minutes on the same graphene sample, with a 96% yield for all devices down to a channel length of 50 μm. The fabricated transistors possess hole and electron mobilities of 663.5 cm^2^/(V·s) and 689.9 cm^2^/(V·s), respectively, and support a simple and effective method of developing high-yield flexible electronics.

## 1. Introduction

Since the 1960’s, integrated circuit electrodes have been fabricated from traditional copper, nickel, silver, and gold metals [1]. Although these traditional metals are used widely in practically all consumer electronics today, they are susceptible to degradation under repeated stress and strain [2]. As the commercial sector demands nanomaterial-based flexible device applications, there is an immediate need for novel and inexpensive fabrication methods that can provide reliable flexible electrodes that resist damage after repeated deformation.

Liquid-Metal (LM) Galinstan is a commercially available eutectic alloy comprising of 68% gallium, 22% indium, and 10% tin that exhibits a conductivity of 2.30 × 10^6^ S/m, a desirable vapor pressure (<1 × 10^−6^ Pa at 500 °C) compared with mercury (0.1713 Pa at 20 °C), and a stable liquid state across a broad temperature range (−19 °C to 1300 °C) [3]. Galinstan has been studied as a candidate for flexible integrated-circuit electrodes due to its deformability and non-toxic nature. Although there are several metals that have higher conductivity compared to LM, such as copper, the low contact resistance of LM in contact with graphene is a desirable benefit over the high contact resistance of metals, such as copper, in contact with graphene [4,5].

To deposit liquid-metal on electronic devices, mask deposition and microcontact printing have been adopted to improve manufacturing yield to over 95%, while simultaneously decreasing feature size down to 1 µm [6,7,8,9,10,11,12]. Such techniques are convenient due to design simplicity, reliability, and high throughput. However, when drop-casting LM with mask deposition, there are immediate drawbacks. The thickness of the patterned LM vary due to the curvature created by the LM surface tension and wetting to the mask and substrate during lift-off. In addition, microcontact printing LM on 2D nanomaterials is not suitable because of the risk of damage to the nanomaterial substrate as the contact head moves or drags across the print surface. A technique known as atomization is desirable for LM deposition on 2D nanomaterials due to the ability to reduce bulk liquids to a fine mist to create thin traces with homogenous thickness. However, the cost of commercial atomization systems is high, making it difficult to adopt such techniques in small labs and standard academic environments [13].

In this article, we demonstrate the use of a low-cost novel spray-on deposition technique for LM using off-the-shelf equipment that improves upon contemporary deposition techniques. We demonstrate the utility of the spray-on deposition of liquid-metal with a proven flexible material combination, consisting of Galinstan source and drain electrodes, an electrolytic gate comprised of honey, and a graphene channel that forms a flexible graphene field-effect transistor [14]. Galinstan was integrated with graphene not only as a solution to enhance flexibility and robustness of the fabricated device, but also as a means to overcome the undesirable high contact resistance of graphene in contact with standard electrode materials copper, gold, and silver [15]. We will aim to convince the reader that our stencil and deposition technique improves yield and simplicity, and decreases cost flexible nanomaterial-based electronics.

## 2. Materials and Methods

Figure 1a–e illustrates the process to fabricate graphene field effect transistors with our low-cost rapid prototyping LM spray-on technique. First, the stencil used to pattern LM electrodes was designed using CAD software (Silhouette Studio, Lindon, UT, USA) and cut into a Polyethylene terephthalate (PET) substrate using a vinyl cutter (Silhouette Portrait, Lindon, UT, USA). The stencil was adhered firmly at the edges with scotch tape onto a commercially bought Chemical Vapor Deposition (CVD) monolayer graphene sample transferred to a PET substrate (Graphene Platform, Shibuya-Ku, Tokyo) (Figure 1a). LM was than loaded into a commercially bought paint-gun reservoir, with the air compressor pressure set to 110 psi. The paint gun was mounted and positioned at a 90-degree angle with the exit aperture facing towards the target surface with an optomechanical stage and sprayed for ~4 s or until a homogenous LM thickness was deposited on the target surface (Figure 1b,c). The stencil was then removed slowly to reveal the desired LM pattern (Figure 1d, Figure 2). This process was repeated to create LM electrode pairs with differing channel lengths. There were four pairs of each of the following channel lengths: 1 mm, 500 μm, 400 μm, 300 μm, 200 μm, 100 μm, and 50 μm (28 electrode pairs in total), on a single 2 inch × 1 inch CVD graphene sample. Our methods are potentially compatible with complex shapes on the order of several hundreds of microns [16]. Out of the 28 electrode pairs, a single 50 μm electrode pair could not be measured. This fabrication error is due to the vinyl cutter reaching its minimum resolution limit. Despite the error, preparation of the LM mask took less than five minutes, and the liquid-metal spray duration took less than 10 s to pattern several pairs of LM electrodes. The total cost of materials, including the graphene sample, is under $200.

It is important to note the quality of monolayer graphene produced commercially has been an issue when fabricating graphene devices, and verification of quality is mandatory before experimentation [17]. The Raman spectrum for the CVD graphene on PET (Graphene Platform) is illustrated in Figure 3. Raman measurements were taken at three different sites and the ratios between the second order overtone (2D) peak and in-plane vibrational mode (G) peak were computed to identify the disorder in graphene [18]. Due to the strong vibrational modes of polymeric PET, the PET Raman spectrum was subtracted from the graphene/PET Raman spectrum, leaving only the Raman spectrum due to graphene. Overall, the samples used for experimentation were of monolayer graphene, with minor disorder due to $I_{2D}/I_{G}\;>2$ for all three spots imaged. There did exist a defect site D, which may be due to the structural disorder caused when transferring graphene to PET. Only electrical measurements can determine the effect the defect site D has on the quality of graphene.

To complete the three-terminal graphene field-effect transistors (GFET) device, honey was loaded into a syringe and drop-casted between each electrode pair to act as an electrolytic gate dielectric, as seen in Figure 4. Honey was chosen as an electrolytic gate dielectric due to its conformability, low-cost, and ease of accessibility [14]. Standard oxides, such as aluminum and silicon oxide, can be used, but are not the focus of this manuscript due to fabrication complexity [19].

## 3. Results and Discussion

To demonstrate the utility of our spray-on LM technique, graphene charge transport characteristics for several GFET devices with varying channel lengths (1 mm, 500 μm, 400 μm, 300 μm, 200 μm, 100 μm, and 50 μm) were extracted from Current-Voltage (I-V) measurements taken via an Agilent 4155C Semiconductor Parameter Analyzer (Santa Clara, CA, USA) and probe station. Each GFET device was subjected to a gate voltage (*V_g_*) sweep from ±5 V with a drain voltage (*V_d_*) of 10 mV and the charge transport characteristics were plotted in Figure 5a,b for comparison. The on:off ratio and gate-leakage current density are also illustrated in Figure 5c and Figure 4d, respectively, for comparison. Tungsten micromanipulator probes were used to make electrical contact to the LM electrodes. Tungsten was chosen as the micromanipulator probe material because tungsten is one of the few materials that does not amalgamate with LM Galinstan [20]. A third micromanipulator probe was used to contact the electrolytic gate dielectric comprised of honey [14] (Figure 4a). Honey was adopted as an electrolytic gate dielectric in order to provide a rapid minimalistic method to actuate the graphene charge transport characteristics. Honey is a polar organic molecule, and in contact with a charged metal surface will form an electric double layer (EDL), as the charged ions that comprise honey diffuse to the graphene/honey interface. Applying either positive of negative potential to the honey via a third micromanipulator probe will enable electron or hole transport in the graphene channel. The authors implore the readers to implement this simple and rapid LM-patterning technique with alternative dielectrics to optimize performance characteristics and tradeoffs for their particular application.

The ambipolar nature of all graphene field-effect transistors is made clear with the V-shape in Figure 5a; *I_ds_* vs. *V_g_* curve. The dual polarity allows the device to operate in either electron or hole conduction mode, which is beneficial for applications such as digital or analog circuit modulation [21]. Notice the minimum drain current, also known as the Dirac peak, for each device is at a negative *V_g_* value, illustrating there is an overall intrinsic n-type doping characteristic that may be brought upon by conduction of electrons through the honey top-gate dielectric. It is commonly known that graphene exhibits p-type behavior in contact with atmospheric oxygen [22]. Therefore, there is reason to believe that the honey allows intrinsic n-type doping behavior and may be due to the composition of sucrose, glucose, fructose, and ash content in the honey [14]. To our benefit, the dirac shift is rather small with respect to *V_g_* = 0 and is rather convenient for low-power devices. This, in part, is due to the nanoscale charge accumulation at the graphene-honey interface, also known as electric double layer (EDL), that can be actuated by altering the gate voltage [23,24,25].

Figure 5c illustrates the on:off ratio for the test devices. With exception to the 100 μm and 50 μm devices, there is a clear trend: as the channel length of each GFET decreases, the on:off ratio increases. This is an expected result based on well-reported improvement of semiconductor devices as miniaturization occurs [26]. In addition to the anomaly in the on:off ratio trend with the 100 μm and 50 μm devices, these devices also exhibit a noticeable asymmetry in electron and hole conduction branches. This asymmetry may be, in part, due to partial charge pinning due to low-resistivity graphene-metal contacts, as has been documented before [27]. However, an observation of the gate current density in Figure 5d shows there is reason to believe the primary cause of asymmetry, and the low on:off ratio, is due to the gate leakage current in short-channel devices. The 100 μm and 50 μm devices possess significant gate current densities in comparison to the other devices. Correspondingly, the 100 μm and 50 μm devices seem to possess the largest asymmetry between electron and hole branch, and exhibit anomalous on:off ratios.

The high gate-leakage current density at the smaller channel lengths may be due to the reduced electrode separation distance between the graphene-dielectric interface. It is noted that in the aforementioned process, the PET stencil did not form an ideal, airtight contact with the graphene in all devices during fabrication. Therefore, there were stray microscale LM-spray residues deposited under the masked areas, and in some cases shorted the source and drain electrodes. In addition, for ease of fabrication, the honey dielectric was drop-casted by hand and applied unnecessary pressure to the LM electrodes causing the LM to move, hence the failure of our methods to produce repeatable working devices at the 50 μm channel length size. Due to these circumstances, there was additional stray conductance between the source/drain electrodes and gate electrode, and the additional gate-leakage current density at smaller channel lengths is to be expected. However, the readers are encouraged to optimize this spray-on process with more ideal stencil-mask materials that provide stronger adhesion to graphene, and produce minimal LM residue. Additionally, other liquid dielectrics, such as lower-conductivity ionic gels, can be utilized in place of honey. Lastly, traditional back-gated graphene field effect transistor topologies can be utilized to eliminate stray conductance produced from overlap of the top-gate dielectric.

The transconductance (*g_m_*) of a 1 mm GFET, Figure 6a, and electron and hole mobilities of the same 1 mm GFET, Figure 6b, were extracted from the following relationships:

Transconductance:(1)$$g_{m}=\frac{\frac{\partial I_{D}}{\partial t}}{\frac{\partial V_{G}}{\partial t}}\;\left(A/V\right)$$

Electron and Hole Mobility:(2)$$\mu_{e},\;-\mu_{p}=\;\frac{Lg_{m}}{WC_{ox}V_{D}}\;({\mathrm{cm}}^{2}/(V\cdot s))$$

The extracted hole and electron mobilities measured for the GFET devices are shown in Table 1 and are comparable to several reported devices with similar methods [14,28,29,30,31,32]. It is important to note the hole and electron mobilities increase as the channel length decreases and may be due, in part, to the lower probability of defects within the graphene channel at shorter channel lengths. However, there was an exception in the 50 μm channel length case and this may be due, in part, to a large dominating gate leakage effect. Other factors may include the quality of the graphene or honey between the LM source and drain electrodes. It is well documented that commercially grown graphene is non-uniform across a given area [33].

## 4. Conclusions

It has been demonstrated that our spray-on LM method is a viable technique to fabricate nanomaterial devices. We have shown that top-gated graphene field-effect transistor devices with feature sizes down to 100 μm can be fabricated and exhibit standard charge transport characteristics with minimal effort. All devices were produced in a time frame of 30 s per transistor with a reliability of 96%. Our methods can be improved with fully-automated spray and drop-casting techniques, or even with back-gated devices, to improve the speed and reliability of this method. Furthermore, a strong-adhesion stencil mask and higher-resolution stencil-cutter can be used to significantly improve the minimum channel length achieved in this first attempt. This technique, with further modification, can be used inexpensively to mass-produce nanomaterial devices.

## Figures and Tables

**Figure 1 micromachines-10-00054-f001:**
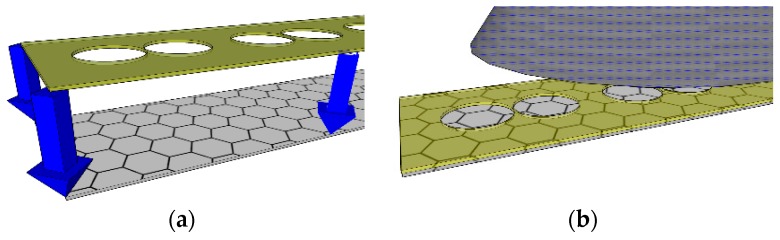
Fabrication process for graphene field-effect transistors with spray-on liquid-metal electrodes: (**a**) electrode patterns are cut into a Polyethylene Terephthalate (PET) substrate and placed flush on graphene surface to act as stencil; (**b**,**c**) liquid-metal is sprayed on the graphene surface with a paint gun; (**d**) the stencil is carefully removed, resulting in patterned electrodes on graphene; (**e**) an electrolytic top-gate material (honey) is drop-casted between the electrode pair to complete the device.

**Figure 2 micromachines-10-00054-f002:**
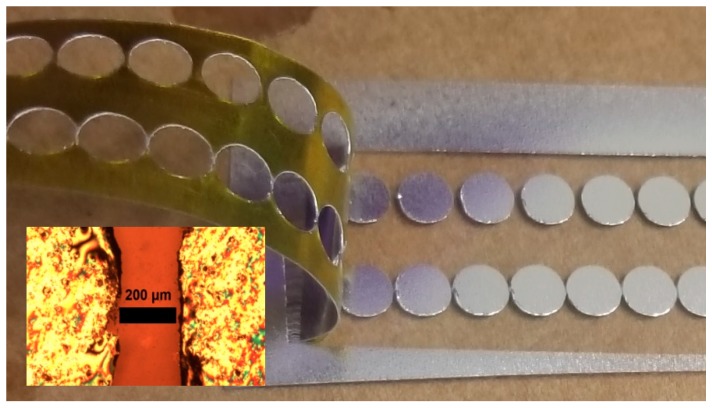
Spray-on Liquid-Metal electrode stencil removal process step. Inset: Magnified view of a liquid-metal electrode pair with a channel width of 200 μm.

**Figure 3 micromachines-10-00054-f003:**
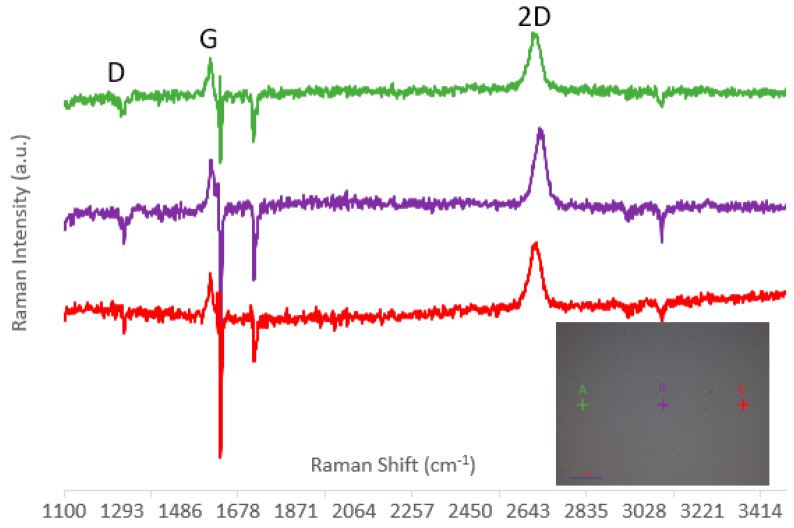
Raman Spectrum of graphene with in-plane vibrational mode (G), second order overtone (2D), and defect site (D) identified. Each Raman measurement was taken with a 532 nm, 2 mW Laser, with an exposure time of 5 s over an area of 25 μm. Inset: Image of Chemical Vapor Deposition (CVD) graphene on Polyethylene terephthalate.

**Figure 4 micromachines-10-00054-f004:**
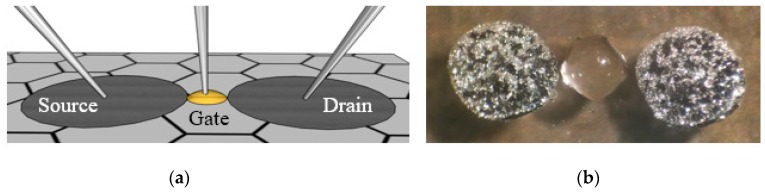
(**a**) Illustration and (**b**) picture of graphene field-effect transistor with spray-on liquid-metal electrodes and honey gate dielectric.

**Figure 5 micromachines-10-00054-f005:**
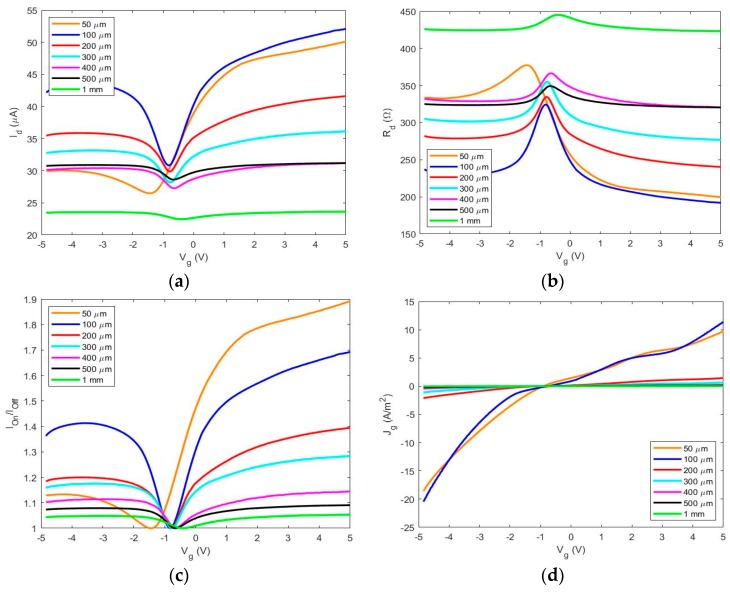
Graphene charge transport characteristics for several graphene field-effect transistors with varying channel-length: (**a**,**b**) illustrates the drain current (*I_d_*) and drain resistance (*R_d_*) as a function of top-gate voltage (*V_g_*); (**c**) details the on:off ratio (*I_on_*/*I_off_*) and (**d**) illustrates the gate-leakage current density.

**Figure 6 micromachines-10-00054-f006:**
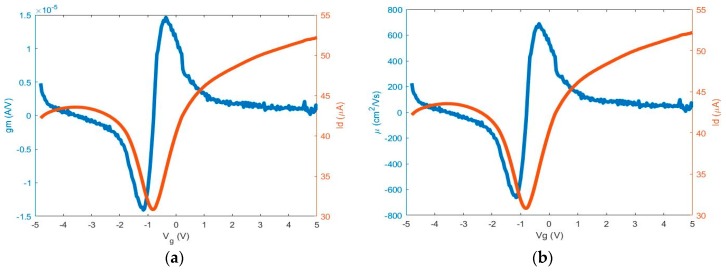
(**a**) Transconductance and (**b**) mobility vs. gate voltage.

**Table 1 micromachines-10-00054-t001:** This table details the electron and hole mobility of each channel length device and is compared to those in other works.

Device	Hole Mobility (cm^2^/(V·s))	Electron Mobility (cm^2^/(V·s))
1 mm	60.17	40.47
500 μm	179.5	105.3
400 μm	241	151
300 μm	371	346
200 μm	393.8	438.5
100 μm	663.5	689.9
50 μm	123	543
Ordonez [14]	213	166
Lu [28]	300	230
Kim [29]	203	91
Wang [30]	N/A	0.04
Kam [31]	154	154.6
Lee [32]	1188	422

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
