# Peer review of "Spray-On Liquid-Metal Electrodes for Graphene Field-Effect Transistors"

_micromachines, 2019, doi:10.3390/mi10010054_

Round 1

Reviewer 1 Report

In this manuscript, the author investigated and summarized the research of recent trend of Graphene field effect transistors. Among them, Author investigated especially FET that be used liquid metal electrode and electrolyte gate dielectric. And this manuscript shows characteristic of devices and novel fabrication process that takes a very short time. And I think this research area is a little impressive for other researchers.

1. wrong sentences

-line 37, In the value of vapor pressure, the multiply symbol is not ‘×’ but alphabet ‘x’.

-line 120, There is no verb in the sentence. ‘Trend’ is a noun not a verb.

-line 221, ‘adsorbed’ have to be used by the passive.

2. In terms of conductivity,

- Galistan has a normal conductivity against other metals (for example Ag = 6.3×107 S/m, Cu = 5.96×107 S/m, Au = 4.1×107 S/m) and it just have conductivity of tungsten level. Conductivity of Galistan is not concerned for high conductivity compared with other highly conductive metals.

- Additionally, in line 109, author refer that Galistan has a conductivity similar to that of copper but conductivity of copper is larger than that of Galistan. Conductivity of copper is ten times larger than conductivity of copper. I think it lacks validity.

 - The conductivity of the sprayed LM electrode, which is considered to be the only novelty of this paper, is 30 kS/m. Is there any way to increase conductivity of electrode? It is considered to be too low conductivity for using as an electrode.

3. I think that the novelty of this paper is considered for a process with 90% reliability rate under rapid fabrication process. But, I think the research about transistor fabricated in this paper is already done in reference 12 and 13. Therefore, I did not think that this paper is not novel. The author referred that the reliability rate of this fabrication process reaches 90%, but the reliability rate is not pretty good compare with other research’s transistor. For proving novelty of this paper, author should show the superiority of the fabrication process speed and device performance compare with reference that of 12 and 13

4. Part 2. Materials and Methods, description of fabrication process is not enough for comprehending the process.

- In figure1, author mentions the polyimide mask as polyimide stencil. But in this paper, polyimide stencil is not composed of any metals. So, I think that the word ‘stencil’ is not proper.

- I don’t know if I understand because of lack of description for fabrication process. When Double-sided polyimide tape is detached from graphene, does not graphene separated from the PET substrate?

- Author mentions that the minimum distance between source and drain electrode that sprayed LM is 50 μm. Why the distance could not be decrease under 50 μm? I wonder that reason and readers may be wonder that reason. So, I think that the author should shows the data why distance under 50 μm is impossible. Because readers may be wonder that reason. In previous studies, distance between drain and source electrode was reduced up to 1 μm. In comparison, I think 50 μm is pretty large.

- Please describe the fabrication process because this paper has a novelty in terms of the fabrication process. Especially, in line 62 and 63, reader could not understand these process because the meaning of sentence is not obvious. What does mean word ‘mat’ and ‘fed’ in line 62,63?

5. Question for part 3.1 Fabrication Qualitative Analysis: this part is described so abstractly so that the readers may be confused.

- Author mentions ‘more uniform electrode pattern’ in line 102, 103. Readers could not understand the word ‘uniform because there are no criteria. Author have to propose the criteria.

- Additionally, micro size morphology is pretty big obstacle in terms of transistor. So, the word ‘uniform’ is not proper this sentence.

- According to line 105 and 106, electrode spraying the LM for 7 seconds is considered for a thick electrode. As the thickness of the sprayed LM increases, the roughness increases. Therefore, the word ‘inversely’ have to be deleted

- Author mentions that the conductivity of the electrode decreases according to increasing of the ejection distance in line 111 and 112. I think this proposal will be a more reliable claim if author show this phenomenon through experimental data. (addition in figure 3)

- Also, supplement a description of how the roughness affects the transistor and data to measure the roughness. Then, this proposal will be a more reliable claim (addition in figure 3)

- Except above two data, figure 3 is meaningless.

6. Figure 4 (b), what is white powder around electrode? I wonder that the white powder is generated inevitably under fabrication process. If the white powder is Galistan, this device is not considered for deliberate device.

7. In figure 6 experiment, the drain voltage value was not mentioned.

8. I think that it need a more specific description in line 220 ~ 228. It is necessary to clarify how the permittivity changes by the electric field and clearly show how it effects the barrier. Readers could no know what effect apply to barrier from this sentence.

9. Lastly, I wonder how calculate the ‘reliability of 90%’.

Author Response

Please see attached Word Document to see the authors notes

Reviewer 2 Report

The paper proposes a spray-on stencil technique to deposit liquid-metal Galinstan electrodes in top-gated graphene field effect transistors (GFETs). The proposed technique is promising for the realization graphene transistors on flexible substrates.

The topic is interesting for technological application. The paper is quite clear, but the content needs some revision.

Here are suggestions and comments that the authors could consider.

Line 127: “A third micromanipulator probe was used to contact the honey gate dielectric, Figure 4.” Is there any special reason to use honey as gate dielectric? The authors could add motivations.

Lines 157-159: “Notice the minimum drain current, also known as the Dirac peak, for each device is at a negative Vg value. This corresponds with an n-type doping profile brought upon by conduction of electrons through the top-gate dielectric.” Usually air-exposed graphene has a p-type behaviour. Are the authors suggesting here that the gate leakage current (conduction of electrons through the top-gate dielectric) is causing the n-doping of graphene? In this case, this is not a real n–doping, but rather an artifact of the measurement. This point should be clarified.

Which is the channel length of the transistors whose characteristics are shown in figure 5? Can the authors add Ig-Vg curves showing the gate leakage? How does the gate leakage affect the drain current modulation shown in figure 5?

The Id current in figure 5 and 6 seems to decrease at high negative Vg voltages. Is the current going to form a double deep as reported in https://doi.org/10.1088/0957-4484/22/27/275702 ? This should be noted and commented.

The asymmetric behavior of the Id-Vg transfer characteristics is usually ascribed to the interaction between contacts and graphene. See for example https://doi.org/10.1088/0957-4484/26/47/475202 or http://dx.doi.org/10.1063/1.4958618. Here is seems to be contributed also by the gate leakage. This should be highlighted and the references added as an explanation.

Lines 203-204: “ These mobilities increase as the channel length decreases with the exception of the 50 μm device. The large gate leakage mentioned above could be the dominating effect for this drop in performance.” The gate leakage has not been mentioned so far. This point should be extended.

Figure 6d: I suggest adding an inset showing the also gate current. It could make more evident the effect of Ig on the drain current. Have the authors any explanation why the gate current density increases dramatically for the 100 and 50 um transistors?

Lines 205-207: “Other factors may include the quality of the graphene or honey between the Galinstan source and drain electrodes. It is well documented that commerically grown graphene is non-uniform across a given area [26].” I concur on the possible presence of defects in graphene. Actually, the authors should better characterize their graphene, reporting for instance a Raman analysis. Moreover, I am not sure of the explanation of the loss of performance observed at 50 um as caused by defects. Defects are more likely included in a long rather than in a short channel device.

Author Response

Please see the following response for the reviewer

Reviewer 3 Report

Authors have reported a low cost rapid liquid-metal Galinstan spray technique to create a uniform electrode pattern. Authors showed I-V characteristics data in support of their investigation. The paper will be an interesting read for the academics. I would suggest accepting this paper as it is.

Author Response

Please see the following response for the reviewer.

Round 2

Reviewer 2 Report

The authors have made significant changes and improvements in their manuscript and have also corrected some technical errors. They have given convincing responses to almost all the questions and comments that I had raised.

The revised version of the manuscript appears technically sounder. The paper can be accepted for the publication in the current form.